Diospyros rhodocalyx Kurz induces mitochondrial-mediated apoptosis via BAX, Bcl-2, and caspase-3 pathways in LNCaP human prostate cancer cell line

Phongsuwichetsak Chayisara 1
Suksrichavalit Thummaruk 2
http://orcid.org/0000-0003-4166-3591 Chatupheeraphat Chawalit 3
Eiamphungporn Warawan 4
Yainoy Sakda 4
Yamkamon Vichanan 1 vichanan.yam@mahidol.edu
1 Department of Clinical Microscopy, Faculty of Medical Technology, Mahidol University , Nakhon Pathom , Thailand
2 Department of Clinical Chemistry, Faculty of Medical Technology, Mahidol University , Nakhon Pathom , Thailand
3 Center for Research Innovation and Biomedical Information, Faculty of Medical Technology, Mahidol University , Nakhon Pathom , Thailand
4 Department of Clinical Microbiology and Applied Technology, Faculty of Medical Technology, Mahidol University , Nakhon Pathom , Thailand
Moreira Daniel
Electronic publication date: 2024 Jul 1
Publication date: 2024
Volume: 12
Electronic Location ID: e17637
Received 2023 Dec 22; Accepted 2024 Jun 5
Copyright: © 2024 Phongsuwichetsak et al.
Copyright year: 2024
Copyright holder: Phongsuwichetsak et al.
License: This is an open access article distributed under the terms of the Creative Commons Attribution License, which permits unrestricted use, distribution, reproduction and adaptation in any medium and for any purpose provided that it is properly attributed. For attribution, the original author(s), title, publication source (PeerJ) and either DOI or URL of the article must be cited.
License URL: https://creativecommons.org/licenses/by/4.0/

Keywords: Diospyros rhodocalyx Kurz, Prostate cancer, Apoptosis, Anticancer activity, LNCaP

Funding: Graduate Scholarship of the Faculty of Medical Technology, Mahidol University This work was supported by the Graduate Scholarship of the Faculty of Medical Technology, Mahidol University. The funders had no role in study design, data collection and analysis, decision to publish, or preparation of the manuscript.

==============================
Background

Prostate cancer (PCa) is one of the causes of death in men worldwide. Although treatment strategies have been developed, the recurrence of the disease and consequential side effects remain an essential concern. Diospyros rhodocalyx Kurz, a traditional Thai medicine, exhibits diverse therapeutic properties, including anti-cancer activity. However, its anti-cancer activity against prostate cancer has not been thoroughly explored. This study aims to evaluate the anti-cancer activity and underlying mechanisms of the ethyl acetate extract of D. rhodocalyx Kurz (EADR) related to apoptosis induction in the LNCaP human prostate cancer cell line.

Methods

Ethyl acetate was employed to extract the dried bark of D. rhodocalyx Kurz. The cytotoxicity of EADR on both LNCaP and WPMY-1 cells (normal human prostatic myofibroblast cell line) was evaluated using MTS assay. The effect of EADR on the cell cycle, apoptosis induction, and alteration in mitochondrial membrane potential (MMP) was assessed by the staining with propidium iodide (PI), Annexin V-FITC/PI, and JC-1 dye, respectively. Subsequent analysis was conducted using flow cytometry. The expression of cleaved caspase-3, BAX, and Bcl-2 was examined by Western blotting. The phytochemical profiling of the EADR was performed using gas chromatography-mass spectrometry (GC-MS).

Results

EADR exhibited a dose-dependent manner cytotoxic effect on LNCaP cells, with IC50 values of 15.43 and 12.35 µg/mL after 24 and 48 h, respectively. Although it also exhibited a cytotoxic effect on WPMY-1 cells, the effect was comparatively lower, with the IC50 values of 34.61 and 19.93 µg/mL after 24 and 48 h of exposure, respectively. Cell cycle analysis demonstrated that EADR did not induce cell cycle arrest in either LNCaP or WPMY-1 cells. However, it significantly increased the sub-G1 population in LNCaP cells, indicating a potential induction of apoptosis. The Annexin V-FITC/PI staining indicated that EADR significantly induced apoptosis in LNCaP cells. Subsequent investigation into the underlying mechanism of EADR-induced apoptosis revealed a reduction in MMP as evidenced by JC-1 staining. Moreover, Western blotting demonstrated that EADR treatment resulted in the upregulation of BAX, downregulation of BCL-2, and elevation of caspase-3 cleavage in LNCaP cells. Notably, the epilupeol was a prominent compound in EADR as identified by GC-MS.

Conclusion

The EADR exhibits anti-cancer activity against the LNCaP human prostate cancer cell line by inducing cytotoxicity and apoptosis. Our findings suggest that EADR promotes apoptosis by upregulating pro-apoptotic BAX, whereas downregulation of anti-apoptotic Bcl-2 results in the reduction of MMP and the activation of caspase-3. Of particular interest is the presence of epilupeol, a major compound identified in EADR, which may hold promise as a candidate for the development of therapeutic agents for prostate cancer.

Introduction

Prostate cancer (PCa) is the predominant malignancy affecting men worldwide, characterized by a substantial mortality rate. Despite the existence of many effective treatments, PCa remains the fifth-leading cause of cancer-related mortality among males (Sung et al., 2021). The therapeutic options for PCa depend on the progression of the disease and interventions, such as radical prostatectomy (RP), radiation therapy (RT), androgen deprivation therapy (ADT), and chemotherapy (Mottet et al., 2021). However, limitations and challenges associated with these treatments, such as PSA relapse after RP, ADT resistance, and the adverse effects of chemotherapy, have been reported (Asmane et al., 2011; Choo, 2010; Omlin et al., 2015). Moreover, the side effects of these therapies, such as urinary complications, erectile dysfunction, and decreased bone minerals, also impact the quality of life (QoL) for patients (Pezaro, Woo & Davis, 2014). Due to the various side effects associated with the PCa treatment and the recurrence of the disease, the study focusing on increased therapeutic efficacy while minimal toxicity to normal cells may potentially enhance both survival and QoL for PCa patients (Sorrentino & Di Carlo, 2023).

Herbal medicines have long been used as plant-based remedies for promoting health and serve as promising sources for discovering cancer drugs, which may offer minimal side effects (Yin et al., 2013). The pharmacological activities of the phytochemicals derived from plants, such as anti-inflammatory, antioxidant, antimicrobial, and anti-cancer properties, have been reported (Dehelean et al., 2021). Additionally, numerous phytochemicals and plant-derived extracts have undergone extensive investigation for various cancers, including PCa. Notably, pomegranates, lycopene, curcumin, green tea, and broccoli have been subjects of clinical trials focused on PCa treatment (Ghosh et al., 2021). Therefore, the considerable pharmacological diversity present in herbs and plant extracts plays an important role in the development of new therapeutic drugs (Cragg & Pezzuto, 2016).

Diospyros rhodocalyx Kurz, belonging to the Ebenaceae family, has been utilized in traditional medicine for various health concerns, including impotence, leucorrhoea, pyorrhoea, parasitic infection, bleeding control, and abdominal pain relief (Mallavadhani, Panda & Rao, 1998; Rativanich & Dietrichs, 1971; Utsunomiya et al., 1998). Recent studies have demonstrated the diverse therapeutic properties exhibited by D. rhodocalyx Kurz. These include its antioxidant effects (Luanchoy et al., 2014), antimicrobial activity (Minsakorn et al., 2021; Theerachayanan, Sirithunyalug & Piyamongkol, 2007), antidiabetic properties (Thengyai et al., 2020), anti-inflammatory effects (Somwong & Theanphong, 2021), and anti-allergic effects (Kraithep, Oungbho & Tewtrakul, 2008). Furthermore, crude extracts derived from D. rhodocalyx Kurz have exhibited cytotoxic effect against the prostate cancer PC3 cells and the breast cancer MDA-MB-231 cells, with IC50 values of 911.22 and 45.08 µg/mL at 72 h, respectively (Chaisanit et al., 2012; Chawalitpong et al., 2012), indicating their potential for cancer therapy. However, the precise anti-cancer mechanism of D. rhodocalyx Kurz has not been comprehensively investigated.

Given the crucial role of apoptosis regulation in cancer treatment (Pfeffer & Singh, 2018; Pistritto et al., 2016), this study aimed to investigate the anti-cancer properties, encompassing the cytotoxicity and apoptosis induction, exhibited by the ethyl acetate extract from D. rhodocalyx Kurz (EADR) on human prostate cancer cell line (LNCaP) and normal human prostatic myofibroblast cell line (WPMY-1). Additionally, phytochemical profiling of EADR was performed using GC-MS. Collectively, these findings are the first study to reveal the potential of EADR as a promising therapeutic agent for PCa, shedding light on its mechanism for apoptosis induction.

Materials and Methods

Reagents and chemicals

CellTiter 96® Aqueous One Solution Cell Proliferation Assay containing a tetrazolium compound [3-(4,5-dimethylthiazol-2-yl)-5-(3-carboxymethoxyphenyl)-2-(4-sulfophenyl)-2H-tetrazolium; MTS] was acquired from Promega, WI, USA. Cell culture media (RPMI 1640 and DMEM) and DPBS were purchased from Cytiva, Logan, USA. Penicillin-Streptomycin and fetal bovine serum (FBS) were obtained from Invitrogen, CA, USA. Propidium iodide solution (PI), dimethyl sulfoxide (DMSO), and RNase A were obtained from Sigma-Aldrich, St. Louis, MO, USA. FITC Annexin V Apoptosis Detection Kit with PI (cat. 640914) was from BioLegend, CA, USA. Rabbit anti-human BAX (D2E11) (cat. 5023), rabbit anti-human Bcl-2 (D55G8) (cat. 4223), rabbit anti-human cleaved caspase-3 (ASP175/5A1E) (cat. 9664), rabbit anti-human β-actin (13E5) (cat. 4970), and secondary anti-rabbit IgG horseradish peroxidase (HRP)-linked antibody (cat. 7074) were purchased from Cell Signaling Technology, Danvers, MA. Other analytical grade chemicals were obtained from RCI Labscan, Bangkok, Thailand.

Culture of LNCaP and WPMY-1 cell lines

The human prostate cancer cell line (LNCaP) and human prostatic stromal myofibroblast cell line (WPMY-1) were purchased from ATCC (VA, USA). LNCaP and WPMY-1 cells were cultured in the RPMI-1640 and Dulbecco’s Modified Eagle’s Medium (DMEM), respectively. All cell lines were supplemented with 10% FBS and 1% Penicillin-Streptomycin. The cells were maintained under standard conditions at 37 °C in a humidified atmosphere containing 5% CO2.

Ethyl acetate extract of D. rhodocalyx Kurz (EADR) preparation

D. rhodocalyx Kurz was procured from Suphan Buri, Thailand. Ethyl acetate served as the solvent for extracting phytochemical components that dissolve both polar and non-polar compounds (Ng, Samsuri & Yong, 2020; Snyder, 1974). Moreover, ethyl acetate exhibits low cytotoxicity towards cells (Abarca-Vargas, Peña Malacara & Petricevich, 2016). A total of 140 g of dried powder underwent maceration in ethyl acetate (0.5 L, 3 times) at 25 °C over 48 h. The extracted solvents were then collected and filtrated through Whatman no. 2 filter paper (Cytiva, Little Chalfont, UK). Subsequently, the filtrate was concentrated by evaporation using a vacuum rotary evaporator at 45 °C to yield crude extract. The percentage yield of crude extracts was calculated based on the weight (g) of the dry extract divided by the weight (g) of plant extract (Engida et al., 2013). Finally, the EADR was dissolved in dimethyl sulfoxide (DMSO) at a stock concentration of 4 mg/mL and stored at −20 °C.

Determination of cytotoxicity by MTS assay

LNCaP and WPMY-1 cells ( 5×103cells/mL) were treated with varying final concentrations of EADR (5, 10, 20, 30, and 40 µg/mL) for 24 and 48 h. The EADR was dissolved in a medium containing DMSO, with a final concentration of 1% DMSO. Cells in the control group were exposed to medium containing 1% DMSO without EADR (vehicle control). According to the manufacturer’s protocol, 20 µL of reagent was added into each well, resulting in a final concentration of 317 µg/mL of MTS in each well. Subsequently, cells were incubated at 37 °C in a humidified atmosphere containing 5% CO2 for 2 h in the dark. The absorbance was measured at a wavelength of 490 nm using a microplate reader (BioTek Instruments, Inc., Winooski, VT, USA). The half-maximal inhibitory concentration (IC50) was calculated using GraphPad Prism version 8.0 (GraphPad Inc., San Diego, CA, USA).

Cell cycle investigation by flow cytometry

LNCaP and WPMY-1 cells ( 5×105cells/mL) were incubated with various concentrations of EADR (5, 10, and 15 µg/mL) for 24 h. Post-incubation, the cells were then washed twice with chilled DPBS and fixed in 70% ethanol for 48 h at −20 °C. Subsequently, the cells were treated with 100 µg/mL of RNase A for 30 min at 37 °C and stained with 50 µg/mL of PI. The cell cycle distribution was determined using a FACSCantroII flow cytometer and further analyzed using the ModFit LTTM software (Verity Software House, Topsham, ME).

Measurement of cell apoptosis by Annexin V-FITC and PI staining

The assessment of apoptosis induction by EADR was conducted through Annexin V-FITC and PI staining. According to the manufacturer’s protocol, briefly, LNCaP and WPMY-1 cells ( 1×105cells/mL) were treated with 5, 10, and 15 µg/mL of EADR for 24 h. Subsequently, cells were harvested, re-suspended in the binding buffer solution, and stained with 5 µL of Annexin V-FITC and 10 µL of PI at room temperature for 15 min in the dark. Then, 300 µL of Annexin-V binding buffer was added. The apoptotic cells were determined using FACSCantroII flow cytometer and analyzed with FlowJo v.10.8.1 software (FlowJo LLC, Ashland, OR, USA).

Mitochondrial membrane potential (MMP) determination by JC-1 staining

The alterations in MMP were assessed with a JC-1 mitochondrial membrane potential detection kit (Biotium, Hayward, CA, USA) according to the manufacturer’s instructions. JC-1 emits either green or red fluorescence depending on the MMP; the green signal indicates depolarized mitochondria, while the red signaling indicates polarized mitochondria. Consequently, the transition from red to green fluorescence served as a reliable indicator of a reduction in MMP (Sivandzade, Bhalerao & Cucullo, 2019). LNCaP cells ( 1×105 cells/mL) were exposed to various concentrations of EADR (5, 10, and 15 µg/mL) for 24 h. Following treatment, the cells were harvested, suspended in a 1X JC-1 reagent working solution, and incubated at 37 °C with 5% CO2 for 15 min in the dark. Subsequently, the excess dye was washed twice using PBS. The MMP was determined using FACSCantroII flow cytometer and data analysis was performed using FlowJo v.10.8.1 software (FlowJo LLC, Ashland, OR, USA).

Western blot analysis

LNCaP cells ( 3×105 cells/mL) were treated with 5, 10, 15, and 20 µg/mL of the EADR for 24 h. Following the harvest process, the cells were washed with PBS. Subsequently, whole-cell lysates were prepared by scrapping the cells into ice-cold RIPA buffer supplemented with a protease inhibitor cocktail (Millipore, Burlington, MA, USA). The lysates from each sample were centrifuged at 14,000 g for 10 min at 4 °C, and protein concentration in the supernatant was determined using a Bio-Rad Protein Assay Dye Reagent Concentrate (Bio-Rad Laboratories, Richmond, CA, USA), following the manufacturer’s instruction. Equal amounts of protein lysate (30 µg) from each sample were separated by using sodium dodecyl sulfate–polyacrylamide gel electrophoresis (SDS-PAGE) on a 15% gel. Mini protein tetra system (Bio-Rad Laboratories, Richmond, CA, USA) was utilized to electrotransfer the separated proteins onto a polyvinylidene fluoride (PVDF) membrane (Merck, Darmstadt, Germany). The blot was then blocked with 5% skim milk in Tris-buffer saline containing 0.1% Tween 20, pH 7.4 under constant rotation for 2 h. Subsequently, the membranes were incubated overnight with monoclonal antibodies against BAX, Bcl-2, cleaved caspase-3, and β-actin at 4 °C. A secondary anti-rabbit IgG horseradish peroxidase (HRP)-linked antibody was used to immunoblot the membranes at room temperature for 90 min. The protein bands were visualized using Immobilon ECL Ultra Western HRP Substrate (Millipore, Stafford, VA, USA), and chemiluminescence was detected by the ChemiDocTM MP Imaging System. The relative intensity of each band was quantified using Image J software (NIH, USA) with normalization to β-actin as a loading control.

Determination of metabolic profiling of EADR by GC-MS

To determine the phytochemical components present in EADR, 10 mg/mL of EADR was filtrated through a 0.45 µm membrane filter and dried by blowing nitrogen gas until the extract was fully dried. Subsequently, 300 µL of acetonitrile (Millipore, Stafford, VA, USA) was added to the dried extract and transferred to GC vials with inserts. The GC-MS profiling was performed using an Agilent 7010B GC/TQ with double HP-5MS UI capillary column (15 m long, 250 µm internal diameter, 0.25 µm file thickness). The injector, transfer line, and detector were maintained at 280 °C. The quadrupole mass spectrometer scanned across the range of 40–350 with the ionizing voltage of 70 eV. and ion source temperature of 300 °C. The GC operating conditions were set as follows: initial temperature 60 °C, 1 min isothermal, ramp at 30 °C min−1 up to 150 °C, and ramp at 5 °C min−1 up to 310 °C. Helium was utilized as the carrier gas at a flow rate of 1 mL/min and analysis was performed in spitless mode. The GC-MS system was controlled by the Enhanced MassHunter. Identification of phytochemical compounds was conducted by comparative analysis the spectrum of unknown compounds with those in the National Institute of Standard and Technology version 2020 (NIST20) and Wiley Search Libraries 11th Edition, 2017 (W11N17) library. The compound name, CAS number, molecular formula, and compound nature of each compound were retrieved from NIST20 and W11N17 libraries. The percentage of each compound was calculated based on its relative peak in the chromatogram (Lalitha et al., 2021). The collective sum of all compound areas was standardized to constitute 100% of the chromatogram area (Mangas Marín et al., 2018). Only compounds with a similarity index of 80% or higher were considered to be well-matched with the databases, indicating accuracy identification and likely presence in the library, as recommended by NIST (Labs, 2023).

Statistical analysis

The statistical analysis was performed using Graph Pad Prism version 8.4.3 (GraphPad Software, San Diego, CA, USA). The findings were presented as the mean ± standard deviation (SD) derived from three independent experiments. The statistical differences between groups were analyzed by one-way analysis of variance (ANOVA) with Dunnett’s multiple comparison test. Statistical significance was defined as a p-value < 0.05.

Results

Cytotoxicity effect of EADR on LNCaP cells and WPMY-1 cells

To investigate the cytotoxicity of EADR on PCa, the viability of LNCaP and WPMY-1 cells post-EADR treatment were observed using MTS assay. As shown in Fig. 1A, the exposure to EADR ranging from 5–40 µg/mL significantly decreased LNCaP cell viability in a dose-dependent manner after 24 and 48 h. The IC50 values for EADR were determined as 15.43 and 12.35 µg/mL at 24 and 48 h, respectively. Exposure of WPMY-1 cells to EADR also exhibited cytotoxicity after 24 and 48 h (Fig. 1B), with IC50 values of 34.61 and 19.93 µg/mL at 24 and 48 h, respectively. The comparative results of the EADR-treated group and the control group are presented in Table 1. Interestingly, at a concentration below 20 µg/mL of EADR treatment for 24 h, no cytotoxicity on WPMY-1 cells was observed, whereas significant cytotoxicity was evidenced in LNCaP cells. This finding suggests that the treatment of EADR at low concentrations only exhibits a cytotoxic effect on prostate cancer cells.

Figure 1 Cytotoxicity effect of EADR on LNCaP and WPMY-1 cells.

The EADR was used at various concentrations to treat (A) LNCaP and (B) WPMY-1 cells for 24 and 48 h of incubation. The results are indicated as the mean ± SD derived from three independent experiments. A p-value less than 0.05 was considered as a statistically significant difference from vehicle control (VC) (*).

Table 1 The comparative results of the EADR-treated group and the control group.

Group	Conc.
(µg/mL)	% Cell viability of LNCaP	p-value	Group	Conc. (µg/mL)	% Cell viability of WPMY-1	p-value	
24 h								
Control	0	93.65 ± 6.32	–	Control	0	101.9 ± 1.42	–	
EADR	5	82.12 ± 4.50	0.034334	EADR	5	102.9 ± 3.83	0.992577	
	10	67.79 ± 3.06	0.000001		10	98.70 ± 2.35	0.587263	
	20	36.74 ± 3.18	0.000001		20	92.93 ± 2.27	0.015264	
	30	16.15 ± 5.25	0.000001		30	65.77 ± 4.26	0.000001	
	40	11.33 ± 4.01	0.000001		40	28.54 ± 3.46	0.000001	
IC50		15.43 ± 0.59				34.61 ± 0.88		
48 h								
Control	0	93.50 ± 7.20	–	Control	0	103.4 ± 1.14	–	
EADR	5	80.37 ± 2.11	0.003139	EADR	5	102.8 ± 2.09	0.998283	
	10	58.99 ± 2.75	0.000001		10	89.50 ± 4.20	0.000298	
	20	23.82 ± 2.97	0.000001		20	54.66 ± 4.09	0.000001	
	30	13.17 ± 1.64	0.000001		30	13.13 ± 1.97	0.000001	
	40	4.48 ± 1.36	0.000001		40	12.63 ± 2.20	0.000001	
IC50		12.35 ± 0.48				19.93 ± 0.26		

Apoptosis induction effect of EADR on LNCaP and WPMY-1 cells

The effect of EADR on cell cycle distribution was examined using PI staining and flow cytometry analysis. Results revealed that EADR did not alter the cell cycle distribution in either LNCaP or WPMY-1 cells. Nevertheless, the results demonstrated that EADR significantly increased the percentage of cells in the sub-G1 phase, particularly in LNCaP cells. Specifically, treatment with 15 µg/mL of EADR led to a substantial increase in the sub-G1 population of LNCaP cells, reaching 36.49% (Figs. 2A and 2B). Conversely, there was no significant alteration observed in treated WPMY-1 cells (Figs. 2C and 2D). These findings suggest that EADR has the potential to induce cell death specifically in prostate cancer cells.

Figure 2 Effect of EADR on cell cycle in LNCaP and WPMY-1 cells.

Histogram plots of (A) LNCaP and (B) WPMY-1 cells are presented at various concentrations of EADR. The percentage of the cell population in each phase of the cell cycle following treatment with EADR for (C) LNCaP and (D) WPMY-1 cells is detailed in the quantitative results. The results represent the mean ± SD obtained from three independent experiments. Statistical significance was assigned to p-value below 0.05 (*).

Further evaluation of the apoptosis induced by EADR was conducted using Annexin V-FITC/PI staining combined with flow cytometry analysis. Results showed that the EADR treatment significantly elevated the percentage of apoptotic LNCaP cells, accompanied by a simultaneous decrease in viable cells (Figs. 3A and 3C). Specifically, treatment with EADR at concentrations of 5, 10, and 15 µg/mL resulted in apoptotic cell percentages of 22.58%, 34.8%, and 43.90%, respectively (Fig. 3C). Notably, no significant differences were observed in the number of apoptotic and viable cells in WPMY-1 cells across all concentrations of EADR treatment. The percentages of apoptotic cells were 16.99%, 17.05%, and 18.23% for EADR treatments at concentrations of 5, 10, and 15 µg/mL, respectively (Figs. 3B and 3D). Therefore, these findings collectively indicate that EADR induces apoptosis in prostate cancer cells.

Figure 3 Effect of EADR on apoptosis in LNCaP and WPMY-1 cells.

Dot plots of (A) LNCaP (B) WPMY-1 cells following 24 h of incubation with EADR using Annexin V/PI double staining are provided. These plots illustrate the populations of viable cells (Annexin V-PI-), cells in early apoptosis (Annexin V+PI-), cells in late apoptosis (Annexin V+PI+), and necrotic cells (Annexin V-PI+). Histogram plots show the percentage of (C) LNCaP and (D) WPMY-1 cells in each phase of cell death after EADR treatment. The bars represent the mean ± SD of three independent experiments. Statistical significance was assigned to p-values less than 0.05 (*), indicating a significant difference compared to the control group.

Decrease of the mitochondrial membrane potential (MMP) induced by EADR

Since mitochondria play a crucial role in apoptosis regulation, we investigated the impact of EADR on the alteration of MMP in LNCaP cells to explore its mechanism of action. Our results revealed the shift of JC-1 red (indicative of aggregates) to JC-1 green (monomeric form) in the EADR-exposed cells, as illustrated in Fig. 4A. This shift corresponded to a notable reduction in the JC-1 red/green ratio compared to the untreated control (Fig. 4B), indicating the reduction in MMP within the cells. Upon treatment with EADR at various concentrations of 5, 10, and 15 µg/mL, the JC-1 green fluorescence in cells increased to 6.85%, 14.7%, and 34.5%, respectively (Fig. 4A). Furthermore, the application of carbonyl cyanide m-chlorophenyl hydrazone (CCCP) as a positive control elevated JC-1 green fluorescence to 98.4% (Fig. 4A). Collectively, these findings indicate the impact of EADR on MMP, suggesting its potential involvement in driving apoptosis in prostate cancer cells.

Figure 4 Effect of EADR on MMP in LNCaP cells.

(A) Flow cytometry analysis of LNCaP cells was performed following 24 h of incubation with various concentrations of EADR or 50 µM carbonyl cyanide m-chlorophenyl hydrazone (CCCP) as the positive control, utilizing JC-1 staining. Figures illustrate the population of JC-1 Green (monomer), which transitions from JC-1 Red (aggregation) at the left region. (B) Histogram plots display the JC-1 Red/Green ratio of LNCaP cells post-EADR treatment. The representative results from three independent experiments are shown as mean ± SD. Statistical significance (*) was assigned to p-values less than 0.05 compared to the control group.

Effect of EADR on apoptotic-related proteins

To further elucidate the mechanisms driving EADR-induced cell death, Western blot analysis was conducted to assess the expression levels of crucial markers. These markers include pro-apoptotic BAX, anti-apoptotic Bcl-2, and the effector caspase, specifically cleaved caspase-3. As shown in Figs. 5A and 5B, treatment of LNCaP cells with varying concentrations of EADR (5, 10, 15, and 20 µg/mL) resulted in increased expression of BAX, exhibiting fold changes of 1.89, 1.99, 2.08, and 2.56 relative to the control, respectively. Conversely, EADR treatment led to a decrease in Bcl-2 expression, exhibiting fold changes of 1.04, 1.03, 0.72, and 0.67 compared to the control, respectively. Moreover, the expression of cleaved caspase-3 in LNCaP cells was elevated, demonstrating fold changes of 1.37, 2.61, 6.64, and 13.86 with EADR treatment at concentrations of 5, 10, 15, and 20 µg/mL, respectively. Therefore, these findings substantiate that EADR induces apoptosis by upregulating BAX, downregulating Bcl-2, and consequently activating the caspase-3 pathway in LNCaP cells.

Figure 5 Effect of EADR on apoptotic protein expression in LNCaP cells.

Western blot analysis was employed to assess the expression levels of (A) BAX, Bcl-2, and cleaved caspase-3 in LNCaP cells after exposure to the different concentrations of EADR. The graph represents the relative intensity of (B) BAX, Bcl-2, and cleaved caspase-3 proteins in relation to EADR concentration. The data are presented as mean ± SD of results obtained from three independent experiments. Statistical significance was assigned to p-values less than 0.05 (*), indicating a significant difference compared to the control group.

Conducting phytochemical profiling of EADR using GC-MS

The compound profiling of EADR, along with the biological activities of the identified compounds, is detailed in Table 2. A total of thirty-eight compounds were identified through spectral matching with NIST20 and W11N17, accompanied by their molecular formulas and references to previously reported biological activities. Notably, among the compounds present in EADR, epilupeol, specifically 20(29)-Lupen-3alpha-ol (isomer 2), exhibited the highest relative percentage of the area at 44.8975%, followed by lup-20(29)-en-3-one at 37.5088%.

Table 2 Phytochemical profiling detected in EADR using GC-MS.

S
No.	RT (min)	Name of compound	Relative area (%)	Similarity (%)	Molecular formular	Compound nature	Biological activity	
1	3.48	2(5H)-Furanone	0.0221	93	C4H4O2	Butenolide	– Anti-tumor activity (Wu et al., 2017)

	
2	3.75	Benzaldehyde	0.0172	94	C7H6O	Aromatic aldehyde	– Anti-oxidant activity and anti-microbial activity (Ullah et al., 2015)

– Anti-tumor activity (Ariyoshi-Kishino et al., 2010)

	
3	3.80	Phenol	0.0086	88	C6H6O	Hydroxybenzene		
4	3.96	1H,1H,2H,2H-Perfluorooctan-1-ol	0.6355	82	C8H5F13O	Fluorotelomer alcohols	– Endocrine disrupted activity (Liu et al., 2009)

	
5	4.07	1-Hexanol, 2-ethyl-	0.043	87	C8H18O	Fatty alcohols		
6	4.96	1-Propanone, 1-phenyl-	0.025	93	C9H10O	Aromatic ketone		
7	5.00	Borneol	0.0633	91	C10H18O	Terpene	– Anti-cancer activity (Cao et al., 2019)

– Anti-oxidant activity (Kumar, Kumar & Raja, 2010

	
8	5.54	Dodecane, 4,6-dimethyl-	0.0299	90	C14H30	Alkane		
9	5.56	Benzene, 1,3-bis(1,1-dimethylethyl)-	0.0493	93	C14H22	Alkylbenzene		
10	5.61	Benzaldehyde, 4-methoxy-	0.0142	85	C8H8O2	Benzaldehydes		
11	6.05	Nonane, 4,5-dimethyl-	0.0474	83	C11H24			
12	6.06	2-methoxy-4-vinyl-phenol	0.036	89	C9H10O2	Phenols	– Anti-cancer activity (Kim et al., 2019)

	
13	6.73	Undecane	0.0208	87	C11H24	Alkane	– Anti-allergic and anti-inflammatory activity (Choi, Kang & Park, 2020

	
14	6.83	Benzene, 1,2-dimethoxy-4-(2-propenyl)-	0.0493	94	C11H14O2	Phenylpropanoid		
15	7.84	Hexadecane	0.0937	93	C16H34	Alkane		
16	7.95	Nonadecane	0.0088	85	C19H40	Alkane		
17	8.06	2,4-Di-tert-butylphenol	0.0942	97	C14H22O	Phenols	– Anti-inflammatory activity and anti-cancer activity (Nair et al., 2020)

– Anti-biofilm activity (Padmavathi et al., 2015)

	
18	8.17	2,6-Di-tert-butyl-4-methyl-phenol	0.0049	82	C15H24O	Phenols	– Anti-inflammatory activity (Murakami et al., 2015)

– Anti-biofilm activity (Santhakumari et al., 2018)

	
19	8.30	Benzoic acid, 4-ethoxy-, ethyl ester	0.041	91	C11H14O3	Benzoate ester		
20	8.34	Hexadecane	0.0205	84	C16H34	Alkane		
21	8.86	Tetracosane	0.0105	81	C24H50	Alkane	– Anti-cancer activity (Uddin, Grice & Tiralongo, 2012)

	
22	10.09	Cyclohexanemethanol, 4-ethenyl-α,α,4-trimethyl-3-(1-methylethenyl)-, [1R-(1α,3α,4β)]-	0.0432	83	C15H26O	Sesquiterpenoid		
23	10.09	Agarospirol	0.0337	84	C15H26O	Terpenes		
24	10.28	β-Eudesmol	0.0651	91	C15H26O	Terpenes	– Anti-inflammatory activity (Kim, 2018)

	
25	10.92	Hexadecane	0.0355	88	C16H34	Alkane		
26	11.61	Nonadecane	0.0147	86	C19H40	Alkane		
27	19.09	Octadecane, 1-isocyanato-	0.149	85	C19H37NO	Cyanates		
28	24.78	1,3-Butadiene, 1,4-diphenyl-, (E,E)-	0.0954	81	C16H14	Alkenes		
29	24.83	1,2-Benzenedicarboxylic acid, 1,2-bis(2-ethylhexyl) ester	0.1033	94	C24H38O4	Phthalate ester		
30	24.98	(2,3-Diphenylcyclopropyl)methyl phenyl sulfoxide, trans-	0.1179	82	C22H20OS	Sulfoxide	– Anti-inflammatory activity (in silico docking study) (Deepa, Sujatha & Velmurugan, 2019

	
31	25.36	5,5′-((2R,3S)-2,3-Dimethylbutane-1,4-diyl)bis(benzo[d][1,3]dioxole)	0.0333	85	C20H22O4	Benzodioxole		
32	26.64	4-((2S,3R)-4-(Benzo[d][1,3]dioxol-5-yl)-2,3-dimethylbutyl)-2-methoxyphenol	0.0806	90	C20H24O4	Lignans		
33	30.48	(4R,4aR,6aS,7S,11aS,11bR)-4-(Methoxymethoxy)-4a,7,11a-trimethyl-tetradecahydro-8H-cyclohepta[a]naphthalen-8-one	0.0164	85	C20H34O3			
34	33.83	Bis(2,6-dimethyl-4-methoxyphenyl)-(1-methyl)butylborane	0.0171	88	C23H33BO2			
35	35.90	Lup-20(29)-en-3-one	37.5088	87	C30H48O	Triterpenoid	– Anti-mucus (Yoon et al., 2015)

– Anti-cancer activity (Bednarczyk-Cwynar, Wiecaszek & Ruszkowski, 2016)

– Anti-mucus (Yoon et al., 2015)

– Anti-cancer activity (Bednarczyk-Cwynar, Wiecaszek & Ruszkowski, 2016)

	
36	36.24	Epilupeol; 20(29)-Lupen-3alpha-ol (isomer 2)	44.8975	86	C30H50O	Triterpenoid	– Anti-inflammation (Sánchez-Ramos et al., 2023; Yasukawa et al., 1995)

– Anti-oxidant activity (Romero-Estrada et al., 2016)

– Anti-tubercular activity (Akihisa et al., 2005)

	
37	37.01	Methyl 4-methyl-2-(2′-nitrophenyl)-5-oxo-1,2,5,7-tetrahydrofuro[3,4-b]pyridine-3-carboxylate	0.0254	85	C16H14N2O6			
38	37.24	Methyl 7-oxodehydroabietate	0.0095	82	C21H28O3			
Note:

RT; retention time

Discussion

D. rhodocalyx Kurz has long been utilized in Thai traditional medicine for treating impotence, increasing libido, and promoting longevity (Othong, Trakulsrichai & Wananukul, 2017). This study is the first to investigate the anti-cancer properties of D. rhodocalyx Kurz, encompassing assessments of cytotoxicity and apoptosis induction, particularly against LNCaP prostate cancer cells. Our results revealed a dose-dependent cytotoxicity effect of EADR on LNCaP cells, with IC50 values of 15.43 and 12.35 µg/mL at 24 and 48 h, respectively. A lesser cytotoxicity of EADR was observed on WPMY-1 cells, with the IC50 values of 34.61 and 19.93 µg/mL after 24 and 48 h of exposure, respectively. This finding suggests that low concentrations of EADR treatment exhibit a cytotoxic effect on LNCaP prostate cancer cells while causing minimal cytotoxicity on WPMY-1 normal human prostatic myofibroblast cells. Similar cytotoxic effects against prostate cancer cells have been observed with extracts and isolated compounds from various Diospyros spp. For example, phenolic compounds isolated from the methanolic extract of D. lotus L. including myricetin, quercetin, kaempferol, myricetin-3-O-β-glucuronide, and myricetin-3-O-α-rhamnoside, exhibited the cytotoxicity against LNCaP cells, with IC50 value of 2.1, 0.4, 2.9, 3.0, and 3.2 µg/mL, respectively (Loizzo et al., 2009). Furthermore, diosquinone isolated from the root sample of D. mespiliformis and D. tricolor demonstrated cytotoxicity against LNCaP cells with IC50 of 4.5 µg/mL (Adeniyi et al., 2003).

Currently, targeting cell cycle arrest and apoptosis induction are crucial strategies in cancer therapy, addressing the uncontrolled proliferation and survival mechanisms of cancer cells that evade apoptosis (Hanahan & Weinberg, 2011; Matthews, Bertoli & de Bruin, 2022; Pfeffer & Singh, 2018). Accordingly, extracts and isolated compounds from Diospyros spp. were investigated for their anti-cancer activities, focusing on cell cycle arrest and apoptosis induction. A previous study demonstrated that the 50% ethanol-water extract of D. castanea (Craib) Fletcher twigs induced late apoptosis in HepG2 hepatocellular carcinoma cells (Weerapreeyakul et al., 2016). Furthermore, 24-hydroxyursolic acid, extracted from D. kaki leaves, facilitated apoptosis in HT-29 colorectal adenocarcinoma cells through the modulation of AMPK and COX-2 (Khanal et al., 2010). Notably, the 70% ethanol extracts of the calyx of D. Kaki Thunb. exhibited cytotoxicity against various human colon cancer cells and induced G0/G1 phase cell cycle arrest in HCT116 human colon cancer cells via downregulation of cyclin D1 through phosphorylation-dependent ERK1/2, p38 or GSK3β, and cyclin D1 transcriptional inhibition through Wnt signaling (Park et al., 2017). In contrast to previous studies, our findings demonstrated that EADR treatment did not induce cell cycle arrest in either LNCaP or WPMY-1 cells. It is probable that EADR does not exert an effect on cell cycle regulators, such as cyclins and cyclin-dependent kinases. Instead, EADR treatment significantly elevated the sub-G1 cell population, specifically in LNCaP cells. This rise in the sub-G1 population suggests that EADR may induce apoptosis in prostate cancer cells (Plesca, Mazumder & Almasan, 2008). Consequently, the apoptosis induction of EADR was further investigated by Annexin V-FITC/PI staining and flow cytometry analysis. The results indicated a significant increase in the total number of apoptotic LNCaP cells post-EADR treatment, while no noticeable increase in apoptotic WPMY-1 cells was observed. Consequently, these findings suggest that EADR does not induce cell cycle arrest, it effectively induces apoptosis in LNCaP prostate cancer cells.

Apoptosis is a controlled cellular process to effectively eradicate the damaged cells, resulting in cell death (Elmore, 2007). Within the apoptosis pathway, mitochondrial outer membrane permeabilization (MOMP) plays an important role in regulating the release of pro-apoptotic proteins from mitochondria, ultimately leading to caspase activation and subsequent initiation of apoptosis (Lopez & Tait, 2015). In this study, the reduction of mitochondrial membrane potential (MMP) was investigated to explore the mechanism of apoptosis induced by EADR. Our findings revealed that EADR significantly reduced the MMP in LNCaP cells, potentially contributing to the initiation of apoptosis. Furthermore, the key apoptosis-related proteins were further investigated, including BAX, Bcl-2, and cleaved caspase-3. The results demonstrated that EADR increased BAX expression and enhanced the cleaved caspase-3 while decreasing Bcl-2 expression in LNCaP cells. These observations indicate that EADR induced apoptosis in prostate cancer cells by upregulation of pro-apoptotic BAX levels while downregulation of anti-apoptotic Bcl-2 expression. This molecular modulation led to the subsequent MMP reduction and caspase-3 activation (Harris & Thompson, 2000). Previously, the efficacy of several natural products in decreasing MMP and modulating the expression of BAX, Bcl-2, and caspase-3 have been reported. For instance, the methanolic extract of Artemisia absinthium induced apoptosis in colorectal cancer through MMP destruction and elevation in BAX/Bcl-2 ratio and caspase-3 expression (Nazeri et al., 2020). Similarly, 13-O-acetylsolstitialin A, derived from Centaurea cyanus activated the decrease of MMP and increased BAX/Bcl-2 ratio to induce apoptosis in breast cancer (Keyvanloo Shahrestanaki et al., 2019). Notably, a flavonoid extracted from D. kaki L. leaves exhibited a similar pattern by reducing MMP, suppressing Bcl-2 expression, and enhancing Bax and cleaved caspase-3 levels in PC3 prostate cancer cells, suggesting the induction of apoptosis via the mitochondrial-mediated pathway. Our findings align with these studies, showing that EADR induces apoptosis via the mitochondrial-associated pathway in prostate cancer cells. Furthermore, recent studies have revealed that LNCaP cells are missing the expression of phosphate and tensin homolog (PTEN) due to the presence of one deleted allele and one PTEN mutation (Lotan et al., 2011; Won et al., 2010). This PTEN mutation leads to the increase of the PI3K/Akt/mTOR pathway and overexpression of Bcl-2, promoting prostate cancer survival and therapeutic resistance (Kim et al., 2017; Pungsrinont, Kallenbach & Baniahmad, 2021). Additionally, a previous study has demonstrated that downregulation of Bcl-2 enhances the chemosensitivity in PTEN-mutated prostate cancer cells (Calastretti et al., 2014). Given that EADR downregulates Bcl-2 expression, EADR might potentially induce LNCaP cell death related to PTEN and Bcl-2-associated mechanisms.

The phytochemical screening of the EADR was conducted using GC-MS. The results revealed the presence of 38 compounds, such as 2(5H)-Furanone, Benzaldehyde, Phenol, 2-ethyl-1-hexanol, 1-phenyl-1-Propanone, Borneol, 2-methoxy-4-vinyl-phenol, 2,4-Di-tert-butylphenol, Tetracosane, Lup-20(29)-en-3-one, Epilupeol, and Methyl 7-oxodehydroabietate as detailed in Table 2. Among these compounds, the major ones identified were epilupeol; 20(29)-Lupen-3alpha-ol and Lup-20(29)-en-3-one, both belonging to the lupeol group of pentacyclic triterpenoids, a class of natural products derived from 30 carbon atom precursors (Parmar et al., 2013; PubChem, 2023). Previous studies have shown that triterpenoids are mostly found in several plants such as tea, cannabis, and citrus fruit (Cox-Georgian et al., 2019). Triterpenoids have been extensively reported for their diverse important pharmaceutical activities, such as anti-diabetic, anti-carcinogenic, hepatoprotective, antioxidant, anti-inflammatory, and antimicrobial effects (Bednarczyk-Cwynar, Wiecaszek & Ruszkowski, 2016; Parmar et al., 2013; Xu et al., 2020, 2022). Although the anti-cancer properties of Lup-20(29)-en-3-one and epilupeol; 20(29)-Lupen-3alpha-ol in prostate cancer cells remain underexplored, existing literature has reported the anti-cancer activities of lupeol against prostate cancer cells. Notably, epilupeol, a prominent compound in EADR, has also been identified in the ethanolic extract of D. rhodocalyx Kurz, which has demonstrated cytotoxic effects against PC3 cells (Chaisanit et al., 2012; Somwong & Theanphong, 2021). Previous research into lupeol’s effects has demonstrated significant inhibitory effects on LNCaP cell viability in a dose-dependent manner, along with the induction of apoptosis via the Fas receptor-mediated pathway (Saleem et al., 2005). Moreover, studies have explored the antiproliferative and antimetastatic properties of lupeol and its derivatives against prostate cancer cells, demonstrating their cytotoxicity and ability to inhibit the migration of PC3 and LNCaP cells in a dose-dependent manner (Castro et al., 2019). Additionally, various compounds identified in EADR, such as 2(5H)-Furanone; Benzaldehyde; Borneol; 2-methoxy-4-vinyl-phenol; 2,4-Di-tert-butylphenol; and Tetracosane have also exhibited anti-cancer activity against different cancer types (Table 2) (Ariyoshi-Kishino et al., 2010; Bednarczyk-Cwynar, Wiecaszek & Ruszkowski, 2016; Cao et al., 2019; Kim et al., 2019; Nair et al., 2020; Uddin, Grice & Tiralongo, 2012; Wu et al., 2017). For example, the synthesized 5(5H)-Furanone showed cytotoxicity and G2/M cell cycle arrest in MCF-7 breast cancer cells (Wu et al., 2017). Benzaldehyde had a cytotoxic effect against HCS-2 oral squamous cell carcinoma and induced autophagy (Ariyoshi-Kishino et al., 2010). Borneol synergistically enhanced doxorubicin-induced cytotoxicity and G2/M cell cycle arrest in U251 and U87 glioma cells (Cao et al., 2019).

In summary, the presence of epilupeol; 20(29)-Lupen-3alpha-ol and Lup-20(29)-en-3-one in the EADR extract suggests their potential as compounds with anti-prostate cancer properties. However, further investigations are imperative to elucidate their underlying mechanism of action and assess their therapeutic applications in prostate cancer.

Conclusions

This study provides valuable insights into the potential anti-cancer activities of D. rhodocalyx Kurz against PCa. Our findings revealed the significant cytotoxicity of EADR on prostate cancer cells, while exhibiting minimal effect on normal prostate cells. Interestingly, EADR induced apoptosis in prostate cancer cells by upregulating BAX and downregulating Bcl-2, leading to the subsequent reduction of MMP and activation of caspase-3. Moreover, the GC-MS analysis of phytochemical profiling identified the epilupeol; 20(29)-Lupen-3alpha-ol (isomer 2) and Lup-20(29)-en-3-one were prominent compounds in EADR. As a triterpenoid, these compounds show promise as a candidate for the development of therapeutic agents against PCa.

Supplemental Information

Supplemental Information 1 The raw measurement of results.

Results from MTS assay, cell cycle analysis, apoptosis, mitochondrial membrane potential, and western blot.

Supplemental Information 2 Uncropped Blot.

Additional Information and Declarations

Competing Interests

Author Contributions

Data Availability

The authors declare that they have no competing interests.

Chayisara Phongsuwichetsak conceived and designed the experiments, performed the experiments, analyzed the data, prepared figures and/or tables, authored or reviewed drafts of the article, and approved the final draft.

Thummaruk Suksrichavalit conceived and designed the experiments, performed the experiments, analyzed the data, authored or reviewed drafts of the article, and approved the final draft.

Chawalit Chatupheeraphat conceived and designed the experiments, performed the experiments, analyzed the data, authored or reviewed drafts of the article, and approved the final draft.

Warawan Eiamphungporn conceived and designed the experiments, analyzed the data, authored or reviewed drafts of the article, and approved the final draft.

Sakda Yainoy conceived and designed the experiments, authored or reviewed drafts of the article, and approved the final draft.

Vichanan Yamkamon conceived and designed the experiments, analyzed the data, authored or reviewed drafts of the article, and approved the final draft.

The following information was supplied regarding data availability:

The raw measurements are available in the Supplemental File.

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
