# Peer review of "Diospyros rhodocalyx Kurz induces mitochondrial-mediated apoptosis via BAX, Bcl-2, and caspase-3 pathways in LNCaP human prostate cancer cell line"

_PeerJ, doi:10.7717/peerj.17637_

## Round 0.1 · original submission · Major Revisions

English Quality: A significant concern that requires immediate attention is the quality of the English language throughout your manuscript. The clarity, coherence, and grammatical accuracy of your writing are essential for effectively communicating your research findings. Reviewer 3 specifically advises submitting your manuscript for professional English editing. This is not merely about correcting grammatical errors but also about ensuring that the scientific arguments are presented clearly and logically.

Clarification and Accuracy in Reporting: The most critical issue is the need for clarity and precision in your descriptions, especially concerning the active compound in your extract and its cytotoxic effects. Reviewer 1's comments on making the cytotoxicity results more explicit and adjusting the wording regarding the active compound are crucial for the integrity of your findings.

Methodological Details: The additional methodological details requested, particularly concerning the high DMSO concentration and the specifics of the reagents and protocols used, are essential for the reproducibility and reliability of your results.

Representation of Data in Figures: Ensuring that your figures accurately represent your data is paramount. The concerns raised about the representativeness of the histograms in Figure 2 and the clarification needed in Figure 3 are significant. These figures must accurately reflect the overall results to maintain the validity of your conclusions.

Expansion of Discussion: The manuscript would benefit significantly from a more comprehensive discussion, particularly incorporating more literature comparisons and a deeper exploration of the implications of your findings, as highlighted by both Reviewer 1 and Reviewer 2.


All the points raised by Reviewer 1 should be addressed, as they directly relate to the clarity, accuracy, and robustness of your research findings.

Reviewer 2's suggestions are also crucial, especially the call for a deeper exploration of the existing literature on D. rhodocalyx Kurz and its mechanisms of action.

The major revisions suggested by Reviewer 3, particularly the addition of another prostate cancer cell line and further apoptotic assays, are important for strengthening your study's credibility and should be considered if resources and time permit.

While Reviewer 3’s suggestion for in vivo testing would indeed provide valuable insights, it may be beyond the scope of your current study due to potential resource and time constraints. If in vivo studies are not feasible, consider discussing the potential implications of your findings for future in vivo research.

**Language Note:** The Academic Editor has identified that the English language must be improved. PeerJ can provide language editing services - please contact us at [email protected] for pricing (be sure to provide your manuscript number and title). Alternatively, you should make your own arrangements to improve the language quality and provide details in your response letter. – PeerJ Staff

·

Basic reporting

Largely carried out well. See the comments in 'additional comments' that suggest certain instances where a few English improvements are suggested and more moderated conclusions are included in places, further literature comparisons added in places in the discussion and some example figures shown to be more representative of overall results.

Experimental design

All done well - a couple more details to be added in the Methods, as outlined in the 'additional comments.

Validity of the findings

The link of some conclusions to the results could be strengthened. This is outlined in the 'additional comments'.
Rationale and benefits are clear, data are provided and are largely robust.

Additional comments

Abstract:
-It is not clear from what is written whether there was cytotoxicity in the normal cell line. This should be more explicit.
-It cannot be claimed that the compound in the extract with the highest concentration was the active substituent – this wording should also be altered, perhaps by removing the second to last sentence of the Results and adding ‘...as identified by GC-MS’ to the end of the current last sentence of that section. The last sentence of the conclusion should also be altered for the same reason. It could read: ‘Epilupeol....may hold promise as a candidate...’.

Introduction:
-At the concentrations required for cancer treatment, herbal medicines may show side effects in the same way as any other drug. Add ‘may’ into line 91 to be more accurate: ‘...which may offer minimal side effects...’

Methods:
-Provenance of the cell line is described.
-No comment is made upon the effect of the vehicle control of 1% DMSO. This is a high DMSO concentration, and may be expected to have an effect. Therefore a comment on this should be included.
-Give final concentration of MTS rather than volume added
-Since concentrations of reagents for the Annexin V / PI assay were not given, it would be helpful to add the catalogue number of the kit used.

Figure legends and figures:
-Figure legend grammar can be improved in a few places, as outlined below.

-Figure 1: ‘p is less than 0.05 was considered’...
-Figure 2, line 1, change ‘were’ to ‘are’...’ Line 2, suggested re-phrasing: ‘The percentage of the cell population in each phase of the cell cycle following treatment with EADR of (C)...
-Figure 3: Dot plots of...after 24h of incubation with EADR...’ Need to add this in so it doesn’t look like 24h incubation with the stains themselves. In line 2, swop ‘illustrate’ in instead of ‘illustrating’ and add ‘cells’ at the end of this sentence. The next sentence would be better to read: ‘Histogram plots show the percentage of ... cells in each phase of cell death after EADR treatment.’
-Fix the spelling of apoptosis and necrosis on the dot plots!
-Figure 4: change illustrating to illustrate and same comment with addition of ‘with EADR’.
Add in full name for CCCP, the bar in B. This is in the text.
-Figure 5: make the second sentence into a full sentence by changing ‘Graph representing...’ to ‘graphs represent...’.
-Table 1: Compound 35 is listed twice, with different biological activities. Combine these.

Results:

Figure 2:
-Histograms for LNCaP – a substantial increase in G1 is observed for the 5 and 10 ug/mL treatments, looking like cell cycle arrest occurring. This is not represented in the graph in C, so presumably the histograms chosen were from the one biological replicate that gave around 80% for G1 rather than the 60%-ish of the other two biological replicates. These histograms should be replaced with others than are more representative of the combined data.

Figure 3:
-The unhealthliness of the ‘normal cells’ has not been mentioned, only that there was no change with the treatments. Please add in an acknowledgement of this.
-Add numbers of cells in each quadrant to the dot plots, as was done for the JC-1 staining.
-It is not at all clear here that the dot plots shown represent the results given in the graphs. Once the numbers have been added, please ensure that they do.

Figure 5:
-Why were the results for the 20ug/mL dose, measured by western blotting and shown in the full blots, not reported? This appears to be selective reporting.
-As well, the actin that is shown in the example western is not from the same blot as the example Bax, Bcl-2 or caspase-3 example westerns. This should be rectified, with the matching actin included for each.
-For the caspase-3, 2 of the full blots have 3 bands for the cleaved caspase-3 and 1 of the full blots has only 2 – has this blot been trimmed? A ‘typical’ blot should be included, in this case, one showing all 3 bands.

Discussion:
-This is largely a re-statement of the results. More literature comparisons should be included in the early part of the discussion as well as some more in-depth comments about their own results.
-Line 345 – exchange ‘absent’ for ‘are missing’
-Line 359 mentions a Table 2 which is not present. Should it be Table 1?
-Comment in line 360-362 – the meaning is unclear here. Was a similar compound found? If so, this should be named here.
-Line 363 triterpenoids – plural – are mostly...

Reviewer 2 ·

Basic reporting

The text is written in appropriate English, with careful employment of technically correct terms. Please, check the following lines for typing or English correctness:
- 150 (which dissolved to be 1% DMSO).
- 260 (indicated that EADR induced apoptosis in prostate cancer cells, which leads to cell death.), considering that apoptosis is itself a form of cell death.
- 345, 346 (cells absent the expression of phosphate and tensin homolog (PTEN), but highly express Akt protein).
- Legends of figures 1 and 5 (p < 0.05 were considered).
- In table 2, line 35 is repeated and with different references for biological activity.
The introduction is concise and clear but should explore more deeply what is already known about the anti-cancer activity and mechanism of the Diospyros rhodocalyx Kurz, including details of the study done on prostatic cancer cell line PC-3. This would provide more context on the research field and how the present study can contribute to it.
The structure of the article follows the expected pattern. The figures and tables are relevant to the study and are well-described and labeled. The raw data is available in supplemental files.

Experimental design

Overall, the study is simple and concise, but carefully designed and well-described. Specifically, the provenance of the cell lines used is clearly described.
The research question is well-defined, but a more detailed introduction would clarify how the study can contribute to the research field. For example, a better description of the research done about D. rhodocalyx Kurz on cancer cell lines (lines 107, 108) would highlight the knowledge gap being investigated. The authors cite an article on PC-3 cells, also a prostatic cancer cell line, but give no details about its findings on EADR anti-cancer activities or mechanisms of action.

Validity of the findings

Although the study is carefully designed, with well-described methods and results, most of the discussion is a repetition of the results, with scarce review of the main topics in the literature, such as studies about D. rhodocalyx Kurz on cancer. Moreover, the various compounds of the extract could have different mechanisms of action against cancer, and this should be considered in the discussion. Also, it is not clear if there are EADR anti-cancer mechanisms already reported in the literature, including apoptotic induction, or if the results presented are the first description of such a mechanism. Finally, the phytochemical profiling of EADR by GC-MS could be more explored, including a discussion about the reported biological activities of the characterized compounds, as well as the potential activities of the other compounds in the context of cancer research.

Reviewer 3 ·

Basic reporting

1. The authors are advised to submit this manuscript to a proficient professional specializing in English editing and scholarly writing.
2. The abstract is not concise enough.

Experimental design

This study demonstrates the anti-cancer potential of EADR against the prostate cancer cell line through the induction of cytotoxicity and apoptosis. However, some issues need to be addressed to make further improvements in this study.
1. In this study, the authors used only one prostate cancer cell line to demonstrate the effect of EADR, which has weak credibility. It is recommended that the authors add at least one other prostate cancer cell line (e.g.DU145, PC3) for further investigation.
2. The authors concluded that EADR induces apoptosis. However, besides the flow cytometry analysis, the chromatin morphology of nuclear condensation and fragmentation should be examined using Hoechst 33258 in prostate cancer cells.
3. The clarification of caspase activation has been correlated with the increase of apoptosis. To verify the association between EADR-induced apoptosis and caspase activation, it is advisable to employ the caspase inhibitor Z-VAD-FMK for confirmation purposes.
4. It will be more informative to also test the effect of EADR on prostate cancer cells in vivo. This will make the system more complete.

Validity of the findings

No comment.

---

## Round 0.2 · accepted · Accept

I am pleased to inform you that two of the three reviewers have confirmed that their comments were adequately addressed in the revised manuscript. Although the third reviewer have not submitted a second round of review, they had recommended only minor revisions in the first place.

I have personally reviewed the revised manuscript and am satisfied that all the reviewers' comments have been appropriately addressed.

Based on this thorough review and the satisfactory responses to all comments, I am happy to confirm that this manuscript is now ready for publication.

Reviewer 2 ·

Basic reporting

The revised manuscript “Diospyros rhodocalyx Kurz induces mitochondrial-mediated apoptosis via BAX, Bcl-2, and caspase-3 pathways in LNCaP human prostate cancer cell line” is improved and refined. The issues and concerns listed in my review were properly addressed, especially the discussion section.

In my opinion, the manuscript is suitable for publication in PeerJ.

Experimental design

no comment

Validity of the findings

no comment

Additional comments

no comment

Reviewer 3 ·

Basic reporting

No comment.

Experimental design

No comment.

Validity of the findings

No comment.

Additional comments

No comment.